

# Intercomparability of $X_{CO2}$ and $X_{CH4}$ from the United States TCCON sites

Jacob K. Hedelius[1], Harrison Parker[2], Debra Wunch[3,a], Coleen M. Roehl[3], Camille Viatte[3], Sally Newman[3], Geoffrey C. Toon[4,3], James R. Podolske[5], Patrick W. Hillyard[5,6], Laura T. Iraci[5], Manvendra K. Dubey[2], and Paul O. Wennberg[3,7]

[1]Division of Chemistry and Chemical Engineering, California Institute of Technology, Pasadena, CA, USA
[2]Earth and Environmental Science, Los Alamos National Laboratory, Los Alamos, NM, USA
[3]Division of Geological and Planetary Sciences, California Institute of Technology, Pasadena, CA, USA
[4]Jet Propulsion Laboratory, California Institute of Technology, Pasadena, CA, USA
[5]NASA Ames Research Center, Mountain View, CA, USA
[6]Bay Area Environmental Research Institute, Petaluma, CA
[7]Division of Engineering and Applied Science, California Institute of Technology, Pasadena, CA
[a]Now at Department of Physics, University of Toronto, Toronto, Ontario, Canada

*Correspondence to*: J. Hedelius (jhedeliu@caltech.edu)

**Abstract.** The Total Carbon Column Observing Network (TCCON) has become the standard for long-term column-averaged measurements of $CO_2$ and $CH_4$. Here, we use a pair of portable spectrometers to test for intra-network bias among the four currently operating TCCON sites in the United States (U.S.). A previous analytical error analysis has suggested that the maximum $2\sigma$ site-to-site relative (absolute) bias of TCCON should be less than 0.2 % (0.8 ppm) in $X_{CO2}$ and 0.4 % (7 ppb) in $X_{CH4}$. We find here experimentally that the 95 % confidence intervals for maximum pairwise site-to-site bias among the four U.S. TCCON sites are 0.05–0.14 % for $X_{CO2}$ and 0.08–0.24 % for $X_{CH4}$. This is close to the limit of the bias we can detect using this methodology.

## 1. Introduction

The Total Carbon Column Observing Network (TCCON) is a network of ground based spectrometers that record near infrared (IR) direct solar spectra from which column abundances of greenhouse gases (GHGs) are retrieved (Wunch et al., 2011b, 2015). Column average dry-air mole fractions (DMFs, or



$X_{gas}$ where "gas" is the species of interest) measured by multiple TCCON sites are used to evaluate $X_{gas}$ retrievals from satellite measurements (for example: Dils et al., 2014; Kulawik et al., 2015; Nguyen et al., 2014; Wunch et al., 2011a). TCCON measurements are tied to the World Meteorological Organization (WMO) in situ trace gas measurement scales through extensive comparisons with in situ

DMF profiles obtained by balloon and aircraft measurements (Deutscher et al., 2010; Geibel et al., 2012; Messerschmidt et al., 2011; Washenfelder et al., 2006; Wunch et al., 2010).

For the TCCON to meet the goals of satellite validation and carbon cycle flux studies, measurements need be precise and accurate. Currently, the $2\sigma$ single sounding uncertainties of the TCCON are estimated to be 0.8 ppm (0.2 %) $X_{CO2}$ and 7 ppb (0.4%) $X_{CH4}$ (Wunch et al., 2010). Systematic errors

such as spectral ghosts (Messerschmidt et al., 2010), pressure offsets, instrument misalignment, or improper fitting of the continuum curvature (Kiel et al., 2016) can, however, produce systematic biases between sites that will remain even after averaging many single sounding measurements. An error analysis by Wunch et al. (2015) suggests that biases of 0.2% for $X_{CO2}$ and 0.4% for $X_{CH4}$ could exist in the network even though the retrieval algorithm (GGG) has undergone continual improvements

designed to reduce such biases.

In this study we quantify bias in $X_{CO2}$ and $X_{CH4}$ among the four operational TCCON sites in the United States (U.S.) in 2015. These sites were at 1) the California Institute of Technology (Caltech), Pasadena, California, 2) Armstrong Flight Research Center (AFRC), Edwards, California, 3) Lamont, Oklahoma, and 4) Park Falls, Wisconsin. Bias quantification was accomplished by comparisons with two mobile

EM27/SUN spectrometers (Gisi et al., 2012). A map of the U.S. 2015 TCCON sites is shown in Fig. 1. The campaign is described in Sect. 2, the data processing and some sensitivity tests are described in Sect. 3. Comparisons between the sites are made in Sect. 4.

## 2. U.S. TCCON 2015 intercomparability campaign

This campaign involved simultaneous side-by-side measurements from 2 EM27/SUN instruments with

measurements at the TCCON sites. One EM27/SUN instrument is operated by Caltech and one by Los Alamos National Laboratory (LANL). These instruments have been described in detail elsewhere (Gisi et al., 2012). Briefly, similar to the TCCON spectrometers, they measure direct solar near IR spectra,





albeit at a much lower resolution (0.5 cm$^{-1}$ versus 0.02 cm$^{-1}$). They include an inbuilt solar tracker and are small and stable enough to be easily transported. We also designate them as mFTSs for mobile Fourier Transform Spectrometers herein. For this study, both mFTSs employed the standard InGaAs (Indium Gallium Arsenide) detector. To reduce the potential for drift between the mFTSs, the campaign was completed within a 5 week period. Based on the lack of drift between the two mFTSs, we conclude that the retrievals from the mFTSs' measurements are internally precise over this period so their $X_{gas}$ products can be used as transferable comparison products.

The general strategy of the campaign was to visit each of the 4 TCCON sites shown in Fig. 1 and attempt at least 5 days of measurements. Two mFTSs were used so any drift in the mFTS measurements would be noticed. In addition to the spectrometers, a traveling Coastal Environment Weather Station with a ZENO® data logger and Setra barometer was used for regular meteorological surface measurements at the AFRC, Lamont Oklahoma (OK), and Park Falls Wisconsin (WI) sites. At Caltech the onsite ZENO® data logger and Setra barometer were used. This type of barometer is used at each of the 4 U.S. TCCON sites. The Setra sensor has a resolution of 0.1 hPa and a stated accuracy of 0.3 hPa. A Paroscientific 765-16B Portable Barometric Digiquartz® pressure standard with a stated accuracy of ±0.08 hPa or better was used as a traveling pressure standard. The Digiquartz® was compared with each of the on-site barometers. Surface pressure is important to the $X_{gas}$ retrievals because it is used to derive the pressure-altitude for the site.

In Table 1 we present the dates of the campaign as well as the number of coincident averaged measurements. In some locations, one mFTS recorded significantly fewer spectra than the other due to instabilities in the firmware or software causing acquisition to occasionally halt unexpectedly – these issues were mostly resolved by updating to the latest firmware provided by Bruker$^{TM}$ while at AFRC. Our quality control filters were set after a preliminary look at the data for the $X_{CO2}$ and $X_{CH4}$ ranges observed. For this study our filters included: 392 ppm<$X_{CO2}$<404 ppm, 1.79 ppm<$X_{CH4}$<1.865 ppm, and solar variation<0.5% within an interferogram. Prior to the campaign several of the TCCON sites used a mercury manometer as an absolute pressure reference. In the comparisons shown here, the current version of the public TCCON data (R0 for Park Falls, R1 for all others) are used where the surface pressure measurements at all sites are tied to the Digiquartz® (Iraci et al., 2014; Wennberg et al., 2014a,




2014b, 2014c). The mFTSs used the meteorological data from the Caltech on site station or from the traveling Setra barometer with offsets applied to match the Digiquartz®.

## 2.1 Site characteristics - Caltech

The Caltech site is located in Pasadena, California (34.136°N, 118.127°W, 240 m a.s.l.), in the California South Coast Air Basin (SoCAB). Pasadena is in an urban environment where there are large diurnal variations of $X_{gas}$ pollutants because of emissions and advection (Wunch et al., 2009, 2016). Emissions from the basin are estimated to be 167 Tg $CO_2 \cdot yr^{-1}$ and $448 \pm 91$ Gg $CH_4 \cdot yr^{-1}$ (Wunch et al., 2016). Pasadena is located towards the northern end of the basin which is bounded by mountains. Two additional sides of the basin are also bounded by mountains, and the other side is bounded by the Pacific Ocean. General conditions during the Aug 2015 campaign were mostly clear skies with some cirrus clouds. We treat two different weeks at Caltech separately to estimate the limits of our methodology. The mean measured daytime $X_{H2O}$ for both weeks was $3540 \pm 840$ ppm (1σ).

## 2.2 Site characteristics - AFRC

The Armstrong Flight Research Center (AFRC, also called Dryden, or Edwards) is located in the Mojave desert at 34.960°N, 117.881°W, 700 m a.s.l. It is approximately 100 km north of Caltech and 100 km east of Bakersfield, California. AFRC is on a military base, but the surrounding area is much less densely populated than the SoCAB. The area is mostly flat and devoid of vegetation. General conditions here during the campaign were cloud free with daytime surface temperatures of $36.4^{+4.0}_{-13.2}$ °C (95% CI) and a mean measured daytime $X_{H2O}$ of $2640 \pm 250$ ppm (1σ).

## 2.3 Site characteristics - Lamont

The Lamont, Oklahoma site is located in an agricultural region that is mostly flat with some rolling hills (36.604°N, 97.486°W, 320 m a.s.l.). It is situated on the Atmospheric Radiation Measurement (ARM) Southern Great Plains (SGP) site. The surrounding area is sparsely populated. During the campaign cumulus clouds were present covering from less than 5% to approximately 40% of the sky. The mean measured daytime $X_{H2O}$ for the campaign week was $5080 \pm 890$ ppm (1σ).



## 2.4 Site characteristics – Park Falls

The Park Falls, Wisconsin TCCON site has been described in more detail elsewhere (Washenfelder et al., 2006). Briefly, the site is in a sparsely populated but heavily forested region with low topographic relief (45.945°N, 90.273°W, 473 m a.s.l.). Conditions were highly variable, ranging from nearly cloud

free to full coverage by stratocumulus clouds. Despite planning more days at this site, the often cloudy conditions contributed to collecting the least amount of data. On 11 September 2015, the TCCON IFS 125HR instrument was realigned as part of routine maintenance. We treat the days before and the day after alignment separately. The mean measured daytime $X_{H2O}$ was $2480 \pm 750$ ppm ($1\sigma$) for this time.

## 3. Data Processing and Sensitivity Tests

Parker et al. (2015) reported on the comparability of the mFTSs $X_{gas}$ products during the campaign, and did not report any drift between them. The modulation efficiency (ME) at maximum optical path difference (MOPD) was reported to be 0.997–0.999 for the LANL mFTS throughout the campaign. The reported ME at MOPD for the Caltech mFTS was lower and more variable, though it is unclear whether or not this variation was due to error in the characterization. A combined mFTS comparison product

was created using an unweighted average of the measurements from the two spectrometers based on the recommendations of Parker et al. (2015). This reduces the drift (if any) by one of the instruments. The observed biases of 0.05 ppm $X_{CO2}$ and -1 ppb $X_{CH4}$ between the mFTSs were added to the Caltech mFTS products before combining with the LANL mFTS products.

As a first comparison to the mFTS data, no adjustments to TCCON data are made. These retrievals use

the operational GGG2014 algorithm (Wunch et al., 2015). Retrievals with the mFTSs are also performed using GGG2014 with the EGI (EM27/SUN GGG Interferogram processing) suite for automation purposes (Hedelius et al., 2016). Both the high and low resolution retrievals used the same model pressure, temperature, altitude, and water profiles (pTz+$H_2O$) generated from the NCEP/NCAR 2.5° reanalysis product (Kalnay et al., 1996). One profile interpolated to local solar noon is used per day

in GGG2014.



Several sensitivity tests have already been performed for TCCON retrievals using GGG2014 (Wunch et al., 2015) as well as for the mFTS retrievals using GGG2014 (Hedelius et al., 2016). We repeat some tests for the data collected at the Caltech site. To test the sensitivity to the lower tropospheric temperature, a +10 K change is applied for all levels at or below 700 hPa. The results are shown in Fig. 2 as a function of air mass. We do not expect the temperature sensitivity to be the same for changes over fewer levels. In Table 2 we list changes in $X_{CO2}$ and $X_{CH4}$ at an air mass of 1.5 for temperature changes over different levels.

## 4. Comparisons

Because of different spectral resolutions between the TCCON instruments (0.02 cm$^{-1}$) and the travelling spectrometers (0.5 cm$^{-1}$), we anticipate that there may be systematic differences in their $X_{gas}$ retrievals. Even in the absence of instrumental problems, spectroscopic inadequacies can cause systematic differences that correlate with T (temperature) errors, surface pressure errors, and solar zenith angle (SZA) (Wunch et al., 2011b). In addition, the instruments have different averaging kernels due to differences in spectral resolution. Thus, even though we use the same a priori gas volume mixing ratio (VMR) and temperature profiles, errors therein will produce differences in the retrieved $X_{gas}$ products (e.g. compare Wunch et al., 2015 and Hedelius et al., 2016). In this section we consider five reasons why the $X_{gas}$ products between the two instrument types (mFTSs and TCCON) may differ.

First, we consider air-mass-dependent artifacts that arise due to the effect of spectroscopic errors being resolution-dependent. Second, we consider how surface pressure bias could affect retrievals, noting that surface pressure bias should be minimal amongst the current United States TCCON sites because of standardization to the common traveling Digiquartz® standard. Third we consider effects of errors in the a priori temperature profile on retrievals from higher versus lower resolution spectra. Fourth we consider the effects of differences in sensitivity from the averaging kernels. Finally, we mention how a non-ideal ILS (instrument line shape) may affect retrievals.



## 4.1 Unadjusted comparisons

The comparisons prior to accounting for differences in temperature sensitivities and averaging kernels are shown as boxplots in Fig. 3 ($\Delta$ = TCCON-mFTS). The mFTS data were scaled to match the TCCON product and center the difference about zero, by dividing by scaling factors of 0.9987 for $X_{CO2}$

and 1.0073 for $X_{CH4}$. These factors were based on the TCCON and mFTS data at all sites. We use the convention that the whiskers are 90% confidence intervals (CI).

Air-mass or SZA-dependent differences may arise due to spectroscopic errors (Frey et al., 2015). At higher SZAs sunlight passes through a longer atmospheric path, which increases the depth of the measured transmission lines. Spectroscopic errors can lead to bias that varies with SZA, even in clean

air sites (Wunch et al., 2011b). Though adding in an air-mass-dependent correction did not improve the long-term mFTS to TCCON comparison in previous studies (Hedelius et al., 2016), here we noted significant air-mass-dependencies. Air-mass-dependent corrections are accounted for in TCCON data, but these are developed for the high-resolution observations (Wunch et al., 2011b). When we attempted to correct the $X_{gas}$ from the mFTS measurements as a function of SZA, we noted significant influences

from local sources and sinks, even at the non-Caltech sites. This complicated the separation of the spurious air-mass effects from true atmospheric variation. Additional measurements in areas with little atmospheric variation could aid in accounting for air-mass artifacts (Klappenbach et al., 2015). In this study, we apply a symmetric basis function to the mFTS products following Eq. A12 in Wunch et al. (2011b), with coefficients determined empirically to reduce the overall diurnally-varying difference

data between the mFTS and TCCON retrievals. Further, for estimates of bias we only use data within ±2 hours of local noon so that comparisons are over similar SZAs at all sites.

## 4.2 Surface pressure and temperature considerations

Surface pressure is used in the calculation of the dry air column in GGG. It is an input to the retrievals to set the pressure-altitudes of each site. A +1 hPa bias in surface pressure leads to average biases of

approximately +0.036% $X_{CO2}$ and +0.039% $X_{CH4}$ respectively for $10°<SZA<20°$ and +0.034% $X_{CO2}$ and +0.049% $X_{CH4}$ respectively for $70°<SZA<80°$ (Wunch et al., 2015). All pressure measurements are tied





to the same Digiquartz® sensor, with an accuracy of ±0.08 hPa. Surface pressure errors can therefore be expected to contribute less than 0.01% to the $X_{CO2}$ and $X_{CH4}$ retrievals.

At different temperatures, the distribution of the molecular $J$ states differs, which can affect the relative strengths of overlapping lines from different species. In GGG bands are chosen to be reasonably
temperature insensitive by including both high and low $J$ lines to average out temperature sensitivity. In the lower resolution spectra, lines are less well resolved. When the algorithm attempts to fit the lines, the overall fit may still be good even if fits for individual species are incorrect, but in compensating ways.

We define a temperature error as the NCEP local noon profile temperature interpolated to the surface
minus the measured site temperature at the surface. Histograms of the temperature errors at the different sites are shown in Fig. 4. In general, NCEP temperatures are typically cooler than those measured on site. At AFRC the difference is particularly large: the NCEP reanalysis product underestimates the surface temperatures by ~10 K at times in this desert region for this particular week. We also compared interpolated surface temperatures from the European Centre for Medium-Range Weather Forecasts
(ECMWF, 0.125° × 0.125°), MERRA-2 (Modern Era Retrospective-Analysis for Research and Applications), GEOS-5 (Goddard Earth Observing System Model), and NAM12 (North American Mesoscale Forecast System, 12 km). Model surface temperature is lower than TCCON temperature in all cases, and 3 of the 5 models have noon differences of ~10 K. Differences are ~7 K for GEOS-5 and ~5 K for NAM12. Though error in the measurement may contribute to part of the T difference, the
lower resolution dynamical models may have a difficult time reproducing surface T at AFRC.

To account for error in the a priori temperature profiles near the surface, we apply two different tests separately. First, we define the temperature error from the surface–700 hPa as equal and apply the results described in Sect. 3. Second, we apply corrections defining the temperature error separately at each level. The error at each level $k$ was defined as the difference from the NCEP profile potential
temperature $\theta_{NCEP,k}$ minus the measured $\theta_{measured,s}$ (where $s$ stands for surface) if $\theta_{measured,s} > \theta_{NCEP,k}$ so that potential temperatures aloft are always greater than or equal to the potential temperature at the surface. Both corrections reduce the diurnal trend of the $\Delta X_{CH4}$ and $\Delta X_{CO2}$ during the middle hours of the day, but do not significantly alter the comparisons in the late afternoon. True temperature



profiles are likely different from the NCEP noon profiles. Future releases of GGG will apply a post-facto temperature correction for the lowest 3 km based on temperature dependent water lines (Toon et al., 2016). For future studies, we recommend adding dedicated sondes as part of the instrument suite for these field campaigns.

### 4.3 Averaging kernel differences

Averaging kernels (Fig. 5) are different for the 0.02 cm$^{-1}$ and 0.5 cm$^{-1}$ instruments. We apply Eq. A13 from Wunch et al. (2011a) to the TCCON $X_{gas}$ ($c$) product to reduce the smoothing error (the contribution of different averaging kernels). We denote the mFTS by subscript 1, the TCCON by subscript 2, and the TCCON product adjusted to reduce the smoothing error of the mFTS averaging kernels (AKs) as 1←2.

$$\hat{c}_{1\leftarrow 2} = c_a + (\gamma_2 - 1)\sum_j h_j a_{1j} x_{aj} \qquad (1)$$

A "$\hat{\phantom{x}}$" represents a retrieved quantity, the subscript "a" denotes the prior, $\boldsymbol{h}$ is the pressure weighting function described by Connor et al. (2008), $\boldsymbol{a}$ is the column AK, $\boldsymbol{x}$ is the DMF a priori profile, and $\gamma$ is the overall scaling factor applied to the TCCON a priori profile to obtain the retrieved $X_{gas}$. Both the TCCON and the mFTS use the same a priori profiles. In Eq. 1, the TCCON profile $\gamma \boldsymbol{x}_a$ is treated as an approximation to the true atmospheric DMF profile (compare Eq. 3 from Rodgers and Connor, 2003). This is a better approximation in a sparsely populated location such as Lamont than at Caltech where local anthropogenic emissions strongly influence the atmosphere. However, overall the application of Eq. 1 only makes differences of $0.00^{+0.04}_{-0.04}$ ppm and $0.01^{+0.17}_{-0.07}$ ppb (95% CI) for $X_{CO2}$ and $X_{CH4}$ in this dataset.

GGG a priori profiles do not take into account local anthropogenic emissions at the surface. In Fig. 6 we plot the in situ DMFs of $CO_2$ and $CH_4$ measured near the surface throughout the day as well as those from the a priori profiles used in the GGG2014 retrievals at the Caltech site. The in situ measurements were recorded using a Picarro cavity ring down spectrometer, with standardization by comparison to three NOAA (National Oceanic and Atmospheric Administration) standards every 23 hours. Given the intense local emissions, the measured in situ DMFs are significantly larger than the a prioris near the



surface. Using the same assumptions as Hedelius et al., (2016), the $X_{gas}$ retrievals for 2 instruments in a polluted environment where the true and a priori profiles differ only at the surface are related by:

$$\hat{c}_1 = \frac{a_{1,s}}{a_{2,s}}[\hat{c}_2 - c_a] + c_a \tag{2}$$

Note the error term has been omitted. The subscript s represents the surface. These assumptions are better for $X_{CO2}$ than for $X_{CH4}$ as changes in tropopause height can also make the a priori methane profile

significantly different from the true profile (Saad et al., 2014). Over this time at Caltech, $X_{HF}$ averaged ~50 ppt and $\gamma^{HF}$ averaged ~0.87, suggesting an a priori tropopause height that is too low. Using the β value from Saad et al. (2014) we estimate a 13% difference in $\gamma^{HF}$ due to tropopause height would cause about a 0.24% change in $\gamma^{CH4}$ (~4 ppb), which is large enough that Eq. 2 is not valid for $X_{CH4}$. We apply Eq. 2 to the $X_{CO2}$ TCCON retrievals at the Caltech TCCON site, which leads to an adjustment

of $0.22^{+0.54}_{-0.35}$ ppm (95% CI).

**4.4 Other considerations**

Imperfections in the instrument line shape (ILS) due to misalignment of the FTSs could also cause site biases. The $X_{air}$ parameter from GGG is used as a diagnostic for large misalignments and timing errors. $X_{air}$ is calculated by dividing the sum of all non-water molecules based on the surface pressure by the

retrieved column of dry air based on column $O_2$. Xair should be close to 1.0 and not vary. At Park Falls $X_{air}$ was approximately 0.979 before and 0.983 after alignment. The mean 2% difference reflects errors in the spectroscopic parameters used by TCCON to measure the oxygen column.

**4.5 Biases to overall median**

The medians and standard deviations for data before and after considering differences in AKs, and

surface temperature are shown in Fig. 7. Only data from ±2 hours from local noon are used. We use the Kruskal-Wallis one-way analysis of variance test, which assumes ordinal but not normally distributed data (Kruskal and Wallis, 1952), to compare data from each site to the median of data from all sites. The null of this test is the medians do not significantly differ. Line styles indicate the degree of significance by the Kruskal-Wallis tests.



Pooled differences are listed in Table 3 for three adjustment steps. These are represented by the sum of all the site or group medians from the overall median, as well as the sum in quadrature of the standard deviations of the mean. Park Falls TCCON data prior to realignment of the spectrometer are omitted. The sum of the median differences decreases for $X_{CO2}$ after adjustments. However, this is not true of $X_{CH4}$ which increases in variability after adjustment. Despite this overall increase for $X_{CH4}$, these adjustments better reflect the intercomparability of the sites rather than the intercomparability of measurements from differing instruments. From Table 3, we estimate the average biases of all sites compared to the median to be 0.03% $X_{CO2}$ and 0.07% $X_{CH4}$.

## 4.6 Confidence intervals of the differences

We use the Critchlow-Fligner method to estimate simultaneous confidence intervals (CI) for the differences between all pairs of sites (Hollander et al., 2014). The Critchlow-Fligner test is nonparametric so it is less sensitive to outliers and few assumptions are needed about the distribution of the underlying population of data. We use $\alpha=0.05$ to obtain 95% confidence intervals of the differences between sites. Results are presented in Table 4. This comparison suggests that $X_{CO2}$ at Lamont has a low bias compared with Park Falls and AFRC. Caltech-1 and Caltech-2 appear slightly different, with the Caltech-2 data being lower than Park Falls. AFRC is also lower than Park Falls. The largest difference within a 95% CI is 0.6 ppm between Park Falls and Lamont. However, most mid-range values are ~0.2 to 0.3 ppm. For $X_{CH4}$, Caltech-1 and Caltech-2 do not differ, but are lower than AFRC, Lamont, and Park Falls-1. Lamont is higher than AFRC and Park Falls. The largest difference within a 95% CI is 4 ppb between Lamont and Caltech. Mid-range values are 2–3 ppb.

## 5. Conclusions

We estimate the range of statistically significant site-to-site bias amongst the sites as <0.3 ppm for $X_{CO2}$ and <3 ppb for $X_{CH4}$. These were determined by comparing TCCON data with simultaneously collected data from co-located portable spectrometers, which we have assumed to be internally precise over the duration of the campaign. This assumption is supported by standard deviations of only 0.15 ppm for $X_{CO2}$ and 1 ppb for $X_{CH4}$ for the 10-minute averaged differences between the two mFTS instruments





over the campaign. Five reasons $X_{gas}$ could differ among instruments were considered: differences in averaging kernels, differences in spurious air-mass-dependence from spectroscopy errors, the a priori profile (e.g. temperature profile), instrument misalignments, and measured surface pressure. Of these, only the last three can cause site to site biases in the TCCON, and adjustments to make the mFTS and

5 TCCON datasets more comparable were made to the first three. As the spectroscopy is improved, the data should have smaller air-mass-dependent artifacts. The corrections based on T errors described in Sect. 4.2 are for the differences in sensitivity to T error between the mFTS and TCCON instruments, and not for the different T errors at each TCCON site. Large temperature errors of +10 K from the surface through 850 hPa could cause errors of 0.08 % in $X_{CO2}$ and 0.11 % in $X_{CH4}$ at an air mass of 1.5.

Biases due to a non-ideal ILS will be reduced in future versions of the GGG retrieval algorithm. Biases in surface pressure data can cause site biases, but are expected be less than 0.01% in the current data revisions because surface pressure data were standardized to the same traveling standard. We recommend regular (~annual, depending on the pressure sensor accuracy) comparisons of meteorological pressure measured by onsite barometers with a universal standard for those making

similar column measurements.

Remaining differences are most likely from a combination of other errors mentioned by Wunch et al. (2015), such as instrumental misalignment and Doppler-shifting of solar lines with respect to telluric lines. Some of these uncertainties will be reduced in the next version of GGG. Other remaining differences may be due in part to noise. Sufficiently large sample sizes should have helped reduce bias

from noise, and the 15-minute running standard deviations for TCCON were 0.11% $X_{CO2}$ and 0.13% $X_{CH4}$. Apparent differences between the weeks at Caltech suggest we are near the precision limit of our current methodology. Though we reduced the contributions of $\Delta X_{gas}$ from different instruments, there may remain additional contributions because of differences in resolution (Petri et al., 2012).

United States TCCON site-to-site biases measured herein are within the $2\sigma$ $X_{CO2}$ and $X_{CH4}$ uncertainties

stated by Wunch et al. (2010). We suggest repeat of this study, comparing results from traveling spectrometers with those from the stationary TCCON sites, especially when aircraft and aircore data are not available to check for bias. Others performing similar studies may even consider using three mFTSs so if there is a relative drift from one mFTS it would be noticeable by comparing to the other two. This





can be repeated every few years, or with different sites (e.g. Sha et al., 2016), or with different gases that can be measured with an extended-InGaAs detector with spectral filters (Hase et al., 2016). Similar studies should, however, also consider the current precision limits of these comparisons on various timescales. We hope others will improve on our methodology to estimate inter-site biases using portable

spectrometers. A sufficient number of aircraft profiles may also aide in determining inter-comparability. The NASA Atmospheric Tomography Mission (ATom), for example, will conduct global flights summer 2016 through spring 2018, and will include profile measurements of $CO_2$, $CH_4$, CO, and $N_2O$ over many of the TCCON sites (https://espo.nasa.gov/home/atom). Data from ATom can be used to re-evaluate TCCON uncertainties in the next version of GGG.

**Acknowledgements**

The MODIS Vegetation Indices products were retrieved from the online Data Pool, courtesy of the NASA EOSDIS Land Processes Distributed Active Archive Center (LP DAAC), USGS/Earth Resources Observation and Science (EROS) Center, Sioux Falls, South Dakota, (https://lpdaac.usgs.gov/node/843). The nightlights data products (Version 1 Nighttime VIIRS

Day/Night Band Composites) are generated by the Earth Observation Group, NOAA National Geophysical Data Center (http://ngdc.noaa.gov/eog/viirs/download_monthly.html). Ancillary surface meteorological data at Lamont were obtained from the Atmospheric Radiation Measurement (ARM) Program sponsored by the U.S. Department of Energy, Office of Science, Office of Biological and Environmental Research, Climate and Environmental Sciences Division. ERA-Interim data were

supplied by ECMWF (European Centre for Medium-Range Weather Forecasts, http://www.ecmwf.int/). NAM12 (North American Mesoscale Forecast System [12 km]) data are a product of NOAA (NCEI DSI 6173, gov.noaa.ncdc: C00630).

We thank Gregor Surawicz and Bruker Optics for assistance with the EM27/SUN firmware update, and Matthias Frey for helpful discussions. We thank Heidi Boyden for her assistance at AFRC, and Jeff

Ayers for his onsite assistance at Park Falls. We thank the ARM SGP site for accommodations and Patrick Dowell and Kenneth Teske for their onsite assistance. We thank Chris O'Dell for providing pTz profiles from MERRA-2, and GEOS5.





The authors gratefully acknowledge funding from the NASA Carbon Cycle Science program (grant number NNX14AI60G), and the Jet Propulsion Laboratory. Manvendra K. Dubey acknowledges funding from the NASA-CMS program for field observations and from the LANL-LDRD for the acquisition of the LANL EM27/SUN.

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





**Tables**

| Site | Dates | n TCCON | n CIT mFTS | n LANL mFTS | n Co.[a] |
|---|---|---|---|---|---|
| Caltech-1 | 10 Aug–15 Aug | 708 | 22338 | 18119 | 145 |
| AFRC | 17 Aug–21 Aug | 1831 | 31980 | 22402 | 283 |
| Caltech-2 | 22 Aug–28 Aug | 740 | 26406 | 22382 | 269 |
| Lamont | 31 Aug–4 Sep | 1146 | 31814 | 32454 | 250 |
| Park Falls-1 | 7 Sep–11 Sep | 369 | 14820 | 13746 | 79 |
| Park Falls-2 | 12 Sep | 187 | 6018 | 6130 | 44 |

Table 1. Number of measurements prior to any filtering. [a]Co. = 10 minute averaged two-way coincident mFTS and TCCON data points.



| % change | XCO2 | | | XCH4 | | |
|---|---|---|---|---|---|---|
| | TCCON | mFTS | Δ | TCCON | mFTS | Δ |
| Surf only | -0.004 | -0.008 | 0.005 | 0.005 | -0.043 | 0.048 |
| Surf-925 hPa | 0.026 | 0.014 | 0.012 | 0.039 | -0.074 | 0.113 |
| Surf-850 hPa | 0.084 | 0.066 | 0.018 | 0.110 | -0.093 | 0.203 |
| Surf-700 hPa | 0.141 | 0.128 | 0.013 | 0.171 | -0.177 | 0.347 |

Table 2. Percent changes for T sensitivities at an air mass of 1.5 and a temperature change of +10 K.





| $X_{CO2}$ (ppm) | AM | AM+T | AM+T+AK |
|---|---|---|---|
| $\sum \lvert Md \rvert$ | 0.9 | 0.9 | 0.6 |
| $\sqrt{\sum \sigma^2}$ | 0.8 | 0.8 | 0.8 |
| $X_{CH4}$ (ppb) | | | |
| $\sum \lvert Md \rvert$ | 5.7 | 5.7 | 6.0 |
| $\sqrt{\sum \sigma^2}$ | 4.3 | 4.1 | 4.2 |

Table 3. Pooled differences pre- and post-adjustment for $\pm 2$ hours of local noon. $Md$=median difference from an individual site to the overall median. $\sigma$=standard deviation of measurements at a particular site. Pooled values exclude PF-1. AM=air-mass-adjustment, T=temperature error adjustment, AK=averaging kernel adjustment.





Table 4. Intersite 95% CI differences of differences.

| $X_{CO2}$ (ppm) | CIT-1 | AFRC | CIT-2 | Lamont | PF-1 | PF-2 |
|---|---|---|---|---|---|---|
| **n** | 69 | 85 | 118 | 99 | 31 | 19 |
| **CIT-1** | | [-0.20,0.00] | [-0.29,-0.09] | [-0.38,-0.13] | [0.02,0.32] | [-0.06,0.23] |
| **AFRC** | [-0.00,0.20] | | [-0.19,0.00] | [-0.28,-0.04] | [0.10,0.42] | [0.03,0.33] |
| **CIT-2** | [0.09,0.29] | [-0.00,0.19] | | [-0.19,0.05] | [0.19,0.51] | [0.12,0.44] |
| **Lamont** | [0.13,0.38] | [0.04,0.28] | [-0.05,0.19] | | [0.23,0.61] | [0.15,0.56] |
| **PF-1** | [-0.32,-0.02] | [-0.42,-0.10] | [-0.51,-0.19] | [-0.61,-0.23] | | [-0.29,0.14] |
| **PF-2** | [-0.23,0.06] | [-0.33,-0.03] | [-0.44,-0.12] | [-0.56,-0.15] | [-0.14,0.29] | |
| $X_{CH4}$ (ppb) | | | | | | |
| **CIT-1** | | [0.8,2.1] | [-1.0,0.5] | [2.6,4.1] | [1.2,3.2] | [-0.3,1.6] |
| **AFRC** | [-2.1,-0.8] | | [-2.3,-1.1] | [1.3,2.6] | [-0.1,1.7] | [-1.5,0.0] |
| **CIT-2** | [-0.5,1.0] | [1.1,2.3] | | [2.9,4.3] | [1.5,3.4] | [-0.0,1.9] |
| **Lamont** | [-4.1,-2.6] | [-2.6,-1.3] | [-4.3,-2.9] | | [-2.1,-0.2] | [-3.7,-1.7] |
| **PF-1** | [-3.2,-1.2] | [-1.7,0.1] | [-3.4,-1.5] | [0.2,2.1] | | [-2.6,-0.5] |
| **PF-2** | [-1.6,0.3] | [-0.0,1.5] | [-1.9,0.0] | [1.7,3.7] | [0.5,2.6] | |

Differences for data within ±2 hours local noon after corrections for air-mass, differences in temperature sensitivity errors defining temperature errors layer-by-layer, and a reduction of the smoothing error from different averaging kernels.



Figures

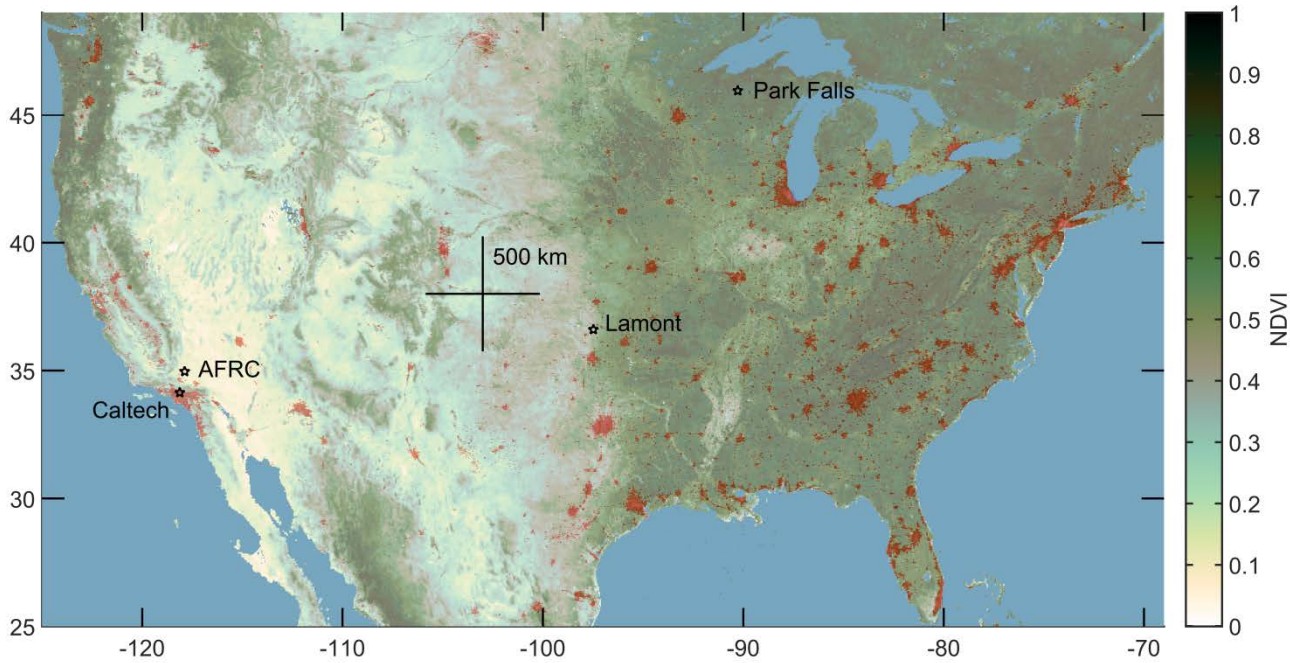

Figure 1. Map of the United States with TCCON sites that were active in 2015 labelled. Normalized
Difference Vegetation Index (NDVI) from Terra MODIS (Moderate Resolution Imaging spectrometer,
Didan, 2015) and nightlights from VIIS (Visible Infrared Imaging Radiometer Suite) in red are shown
for September 2015.





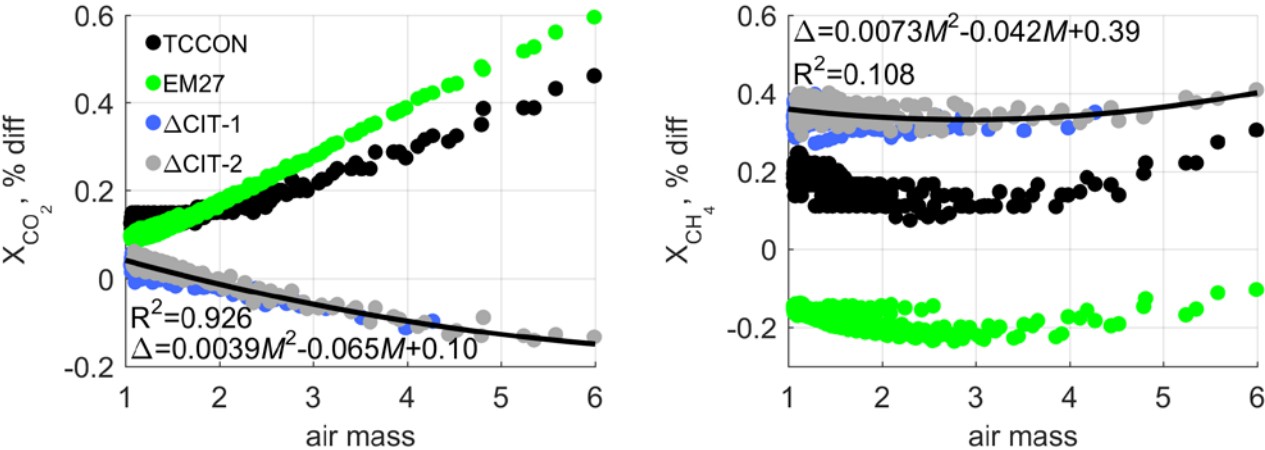

Figure 2. Sensitivity of TCCON and mFTS retrieved $X_{CO2}$ (left) and $X_{CH4}$ (right) to a +10 K change in the PBL (surface–700 hPa) a priori temperature. Green and black points are raw sensitivities, blue and grey points are their differences during the two times at Caltech. Points are 10 minute averages, n=397. For $X_{CO2}$ the TCCON-EM27 differences are small (<0.15%) but air-mass-dependent. For $X_{CH4}$ the TCCON-EM27 differences are larger (0.3-0.4%) but with little air-mass-dependence. The strong air-mass-dependence for $X_{CO2}$ suggests that air-mass needs to be taken into account for $X_{CO2}$ surface temperature error adjustments.





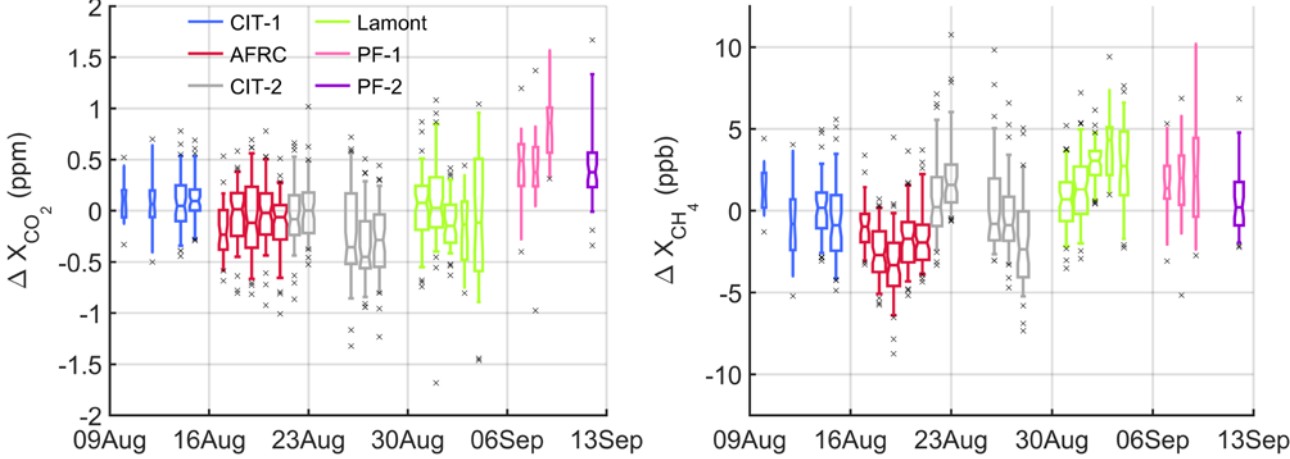

Figure 3. Differences between the TCCON and the mFTS products that are unadjusted except overall scale factors have been applied to the mFTS data ($X_{CO2}$: 0.9987, $X_{CH4}$: 1.0073). Boxplots width represents number of comparison points. They are drawn with the center line as median, the center box is the middle 50% range of data and the whiskers are the 90% range.





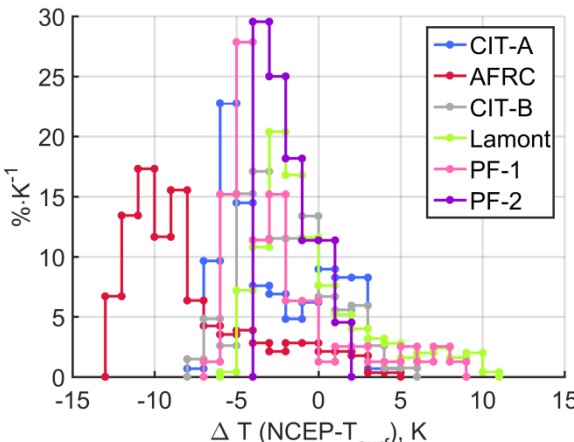

Figure 4. Histograms of differences in temperature from those used in the retrievals at the surface (NCEP model) as opposed to the temperature measured at the TCCON sites.



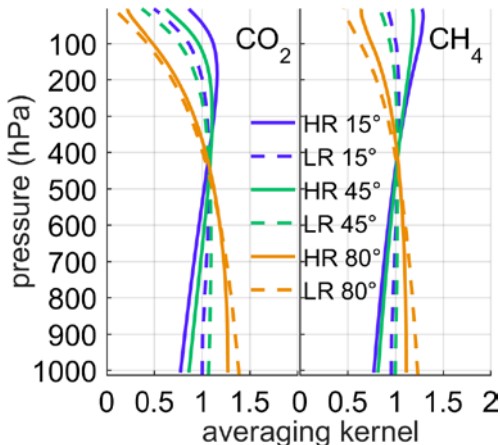

Figure 5. A comparison of the averaging kernels at 3 different SZAs for the high resolution (HR) and low resolution (LR) instruments. The LR instruments are more sensitive to changes at the surface, but less sensitive to changes in the stratosphere.





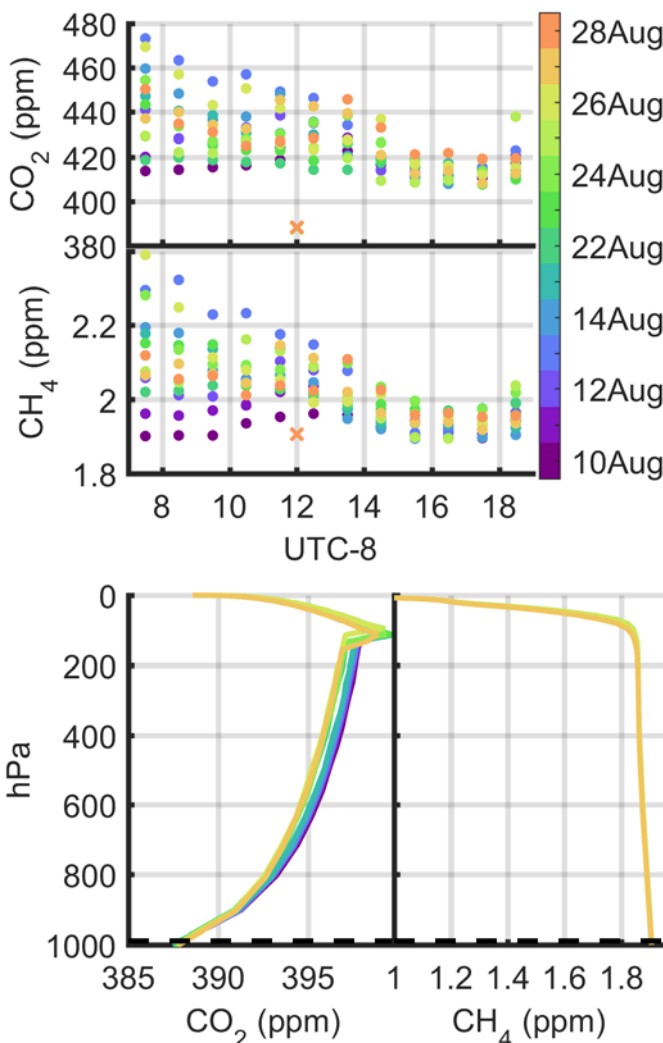

Figure 6. Top panels: diurnal variation of in situ DMFs measured near the surface at Caltech on the days of TCCON to mFTS comparisons. A priori surface values are marked by an "x" at noon. Bottom panels: GGG2014 a priori profiles used in the retrievals, with lower $CO_2$ and $CH_4$ than was measured near the surface. Surface pressure is indicated by the dashed line.





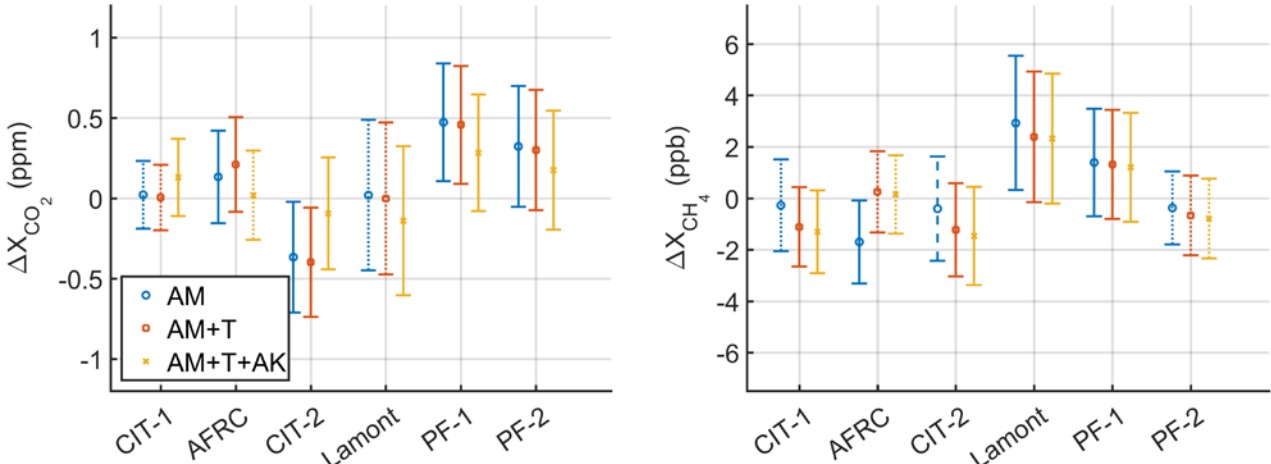

Figure 7. Medians and standard deviations of the TCCON data compared to the mFTS product after various adjustments. Line style represents the significance of the difference of the group median from the median of all data by the Kruskal-Wallis test. ($p<0.05$ –, $p<0.2$ --, otherwise ⋯). Legend entries indicate what adjustments were applied to the data to make measurements from the different instrument types more comparable. AM = air-mass adjustment, T = temperature error adjustment, AK = averaging kernel adjustment.