# Peer review of "Intercomparability of $X_{CO2}$ and $X_{CH4}$ from the United States TCCON sites"

_Atmospheric Measurement Techniques, 2016_

## Referee Comment (RC1) · F. Hase (Referee) · 12 Oct 2016

The paper under consideration investigates the intercomparability of US TCCON sites using the mobile EM27/SUN spectrometer as a travel standard. This is important work in itself as well as in prospect, given the fact that there are TCCON sites located on remote islands or within megacities, where a comparison with in-situ profile measurements collected during aircraft overflights or using aircore balloon technique is difficult or impracticable. In such cases, the mobile spectrometer opens up the opportunity of demonstrating the intercomparability of the remote TCCON site with respect to another TCCON site or a set of TCCON sites used as reference. This work corroborates earlier findings concerning the excellent stability of the mobile spectrometers. The authors perform a sound investigation of the residual differences and of their possible causes. Unfortunately, in my opinion the current version of the manuscript falls short

in delivering what could be an exhaustive investigation. The authors could come up with more definite conclusions concerning the underlying instrumental reasons for the remaining discrepancies based on their observations and monitoring capabilities already implemented in TCCON. In my feeling, it is a pity that the authors stop before that point. I rate the paper excellent and recommend publication, but would urge the authors to incorporate appropriate extensions in the final AMT paper (for this reason, I suggest "major revisions" despite of my positive ranking of the paper in all categories). The authors correctly identify the several problems which emerge from the fact that the spectral resolution of the TCCON observation differs from the spectral resolution of the EM27/SUN. The TCCON data product is derived from a high-resolution spectrum, which cannot be achieved with the EM27/SUN, and therefore the two kinds of observation systems are intrinsically different. This fact evokes subtle differences in the derived column-averaged abundances. The associated uncertainties are taken into account by the authors in an appropriate manner. This is how far one can get in verifying the TCCON results with the low-resolution mobile spectrometers and it is the point where the investigation stops. However, it would, in my feeling, be of substantial interest to enlighten a bit further by which instrumental contributing factors the residual discrepancies are generated. For this purpose, the TCCON interferograms could be truncated to the EM27/SUN resolution. This procedure would generate identical observation systems to a degree that a direct intercomparison of derived mole fractions can be performed. Moreover, it would essentially remove the modulation efficiency variations along optical path difference of the high-resolution spectrometer, which in turn impact the TCCON results (whereas other error contributions, e.g. residual nonlinearity, sampling ghosts, etc, are preserved). Based on this additional data set, the participating TCCON sites could be evaluated twice: firstly, based on the TCCON data products in comparison to the EM27/SUN products, secondly, based on the data products derived from resolution-reduced spectra in comparison to the EM27/SUN products. If the biases found between TCCON sites differ, we would assume that this difference is mainly due to ILS differences between sites. TCCON has implemented an instrumental

line shape (ILS) monitoring based on calibrated gas cells: the authors could therefore check whether the empirical statistical findings are in agreement with the expected error propagation of an imperfect ILS - thereby closing the circle. Note that the foreseen capability of handling an imperfect ILS in the analysis is not required for this investigation - estimates of the sensitivity of TCCON data products with respect to modulation efficiency imperfections have been provided by e.g. D. Griffith and other investigators in the past. However, the exercise will provide a preview of the level of intercomparability which can be expected for TCCON when the ILS biases will be incorporated in the standard TCCON workflow (as announced by the authors).

Minor comments: Page 12, line 1 ff: "Of these, only the last three can cause site-to-site bias". Is this true? If e.g. two TCCON sites are operated in different latitudes, then a comparison between datasets of actual values and TCCON data would reveal a systematic bias between the sites due to spurious air-mass dependence from spectroscopic issues (if we assume that the two sites systematically cover different ranges of solar elevation angles).

Page 12, line 26: The suggestion of using several mobile spectrometers for the intercomparison seems to imply a substantial effort (if the spectrometers are not collocated at the site for performing differential measurements anyway). I would envisage a different recommended standard procedure, especially for remote TCCON sites: the use of a single spectrometer, including a careful demonstration that no instrumental drift occurred (perform an intercomparison with respect to a reference - ideally, a TCCON spectrometer and one or several mobile spectrometers remaining there - before and after the campaign). In this context, I would find it useful to discuss in more detail the level of stability of the participating mobile spectrometers as reaction to transport events. (The LANL spectrometer has been operated site-by-site to the TCCON spectrometer located in Karlsruhe before overseas shipment - it would be interesting to include these observations as well.)

---

## Referee Comment (RC2) · Anonymous Referee #2 · 25 Dec 2016

General comments:

This manuscript presents a study of the intra-network bias of the XCO2 and XCH4
among the four currently operating TCCON sites in the United States using a pair of
portable low resolution spectrometers of the type EM27/SUN designated as mFTSs in
the manuscript. Both, the TCCON spectrometer with a spectral resolution of 0.2 cm-1
and the mFTSs with a spectral resolution of 0.5 cm-1 measure the direct solar radia-
tion in the near infrared spectral region using standard InGaAs detectors. The inter-
comparison campaign was performed within a short time period of 5 weeks to reduce
the potential drift between the mFTSs. The authors consider different reasons for the
residual differences in the Xgas products between the TCCON sites and the mFTSs.
Some of these reasons, like the air-mass-dependent artifacts, surface pressure bias,
a priori temperature profile error and the sensitivity of the averaging kernel difference,

are discussed in detail. However, the manuscript mentions only the possible biases due to the non-ideal instrumental line shape (ILS) without going into further details. In my opinion, this is an important piece of information which would help in understanding the cause of the residual biases between the TCCON sites. The paper is very well written and has a good structure. I recommend its publication after incorporating the following changes.

Major comments:

I suggest to include a plot (or to give values) showing the ILS parameters of the 4 TCCON sites. The sensitivity of the TCCON retrieval results with respect to the ILS parameter has been studied in the past by several investigators. Therefore, the authors could use this information and check if their findings are in agreement with the expected bias due to an imperfect ILS of the TCCON spectrometer.

Furthermore, as correctly pointed out in the manuscript, there exist some problems due to the difference in the resolution of the two spectrometer types. As a result, I suggest performing an additional step of truncating the TCCON measurements to the resolution of the mFTSs and making the intercomparison study. This would create identical measurements performed by the two different types of spectrometers at the same spectral resolution. The inter-comparison of the Xgas retrieved from the two datasets would eliminate several differences which are present when comparing the mFTS retrieval results with the high resolution TCCON retrieval results, but will preserve other errors which are instrument and detector related. This study would therefore give a more exhaustive investigation of the residual bias and their possible causes.

Minor comments:

Page 7 line 4-5: What is the motivation of using a different scaling factor than that of the TCCON Xgas calculation scaling factor?

Technical comments:

Page 6 line 10: Reference missing, e.g. Petri et al. 2012, Gisi et al. 2012

---

## Author Comment (AC1) · 30 Jan 2017

**Responses to referee comments for:**

**Intercomparability of $X_{CO2}$ and $X_{CH4}$ from the United States TCCON sites**

Jacob K. Hedelius, Harrison Parker, Debra Wunch, Coleen M. Roehl, Camille Viatte, Sally Newman, Geoffrey C. Toon, James R. Podolske, Patrick W. Hillyard, Laura T. Iraci, Manvendra K. Dubey, and Paul O. Wennberg

We thank Frank Hase and the anonymous referee for their helpful comments. Their comments have helped us improve this paper.

Our responses are below and are structured as follows: (1) **referee comments are bold, black, and in a sans serif font,** (2) our responses are black and also in a sans serif font, (3) changes to the manuscript are blue and in a serif font.

**Responses to referee Frank Hase**

**1.1**

**The authors could come up with more definite conclusions concerning the underlying instrumental reasons for the remaining discrepancies based on their observations and monitoring capabilities already implemented in TCCON... it would, in my feeling, be of substantial interest to enlighten a bit further by which instrumental contributing factors the residual discrepancies are generated. For this purpose, the TCCON interferograms could be truncated to the EM27/SUN resolution. This procedure would generate identical observation systems to a degree that a direct intercomparison of derived mole fractions can be performed. Moreover, it would essentially remove the modulation efficiency variations along optical path difference of the high-resolution spectrometer, which in turn impact the TCCON results (whereas other error contributions, e.g. residual nonlinearity, sampling ghosts, etc, are preserved). Based on this additional data set, the participating TCCON sites could be evaluated twice: firstly, based on the TCCON data products in comparison to the EM27/SUN products, secondly, based on the data products derived from resolution-reduced spectra in comparison to the EM27/SUN products.**

This suggestion to compare truncated 125HR and mFTS measurements is excellent. Such a comparison would eliminate many possible discrepancies from comparing results from different resolution instruments. We have run GFIT retrievals on spectra generated from truncated interferograms obtained from the 125HR instruments. In the new section 4.5 we discuss the results from truncating the 125HR interferograms. We have also added the results to Table 3, and Figures 9 & 10. The text from Sect. 4.5, Table 3 and Figures 9 & 10 are copied below.

**4.5 Truncated 125HR interferograms comparisons**

Retrievals from the 125HR and mFTS instruments are inherently different due to the differences in resolution. By truncating the longer 125HR interferograms to the same length as those collected from the mFTS, similar resolution spectra are obtained. This likely eliminates most discrepancies between the different types of measurements, except for some residual instrumental imperfections such as instrument misalignment or ghosts. Truncation also reduces the effects of ME variations due to the smaller MOPD. Truncation has been performed in past studies comparing retrieved $X_{gas}$ from different resolution spectrometers (Gisi et al., 2012; Hedelius et al., 2016; Petri et al., 2012). This test provides little new information if truncation changed all retrieved DMFs in a uniform manner. However, past studies showed truncation does not necessarily affect all results the same way, which makes this test imperative in diagnosing potential causes of differences. It helps in determining which biases likely arise from instrumental issues, and which arise from other issues such as errors in the forward model (e.g. from temperature biases at different locations).

The results of the truncation test are shown in Fig. 9, and changes are most easily seen from the unscaled (open) points. The sign of the change for $X_{CO2}$ is both positive and negative for the different sites. Previous studies also noted changes that were negative (Petri et al., 2012), positive (Gisi et al., 2012), or both (but with a preference towards negative) (Hedelius et al., 2016) when using lower resolution spectra. For lower resolution spectra $X_{CH4}$ increases, in agreement with previous studies (Hedelius et al., 2016; Petri et al., 2012).

Table 3. Mean differences pre- and post-adjustment for ±2 hours of local noon.

| $X_{CO2}$ (ppm) | AM | AM+T | AM+T+AK | Trunc |
|---|---|---|---|---|
| $\frac{1}{n}\sum\|Md\|$ | 0.17 | 0.18 | 0.11 | 0.14 |
| $\frac{1}{n}\sum\|\sigma\|$ | 0.34 | 0.34 | 0.34 | 0.42 |
| $X_{CH4}$ (ppb) | | | | |
| $\frac{1}{n}\sum\|Md\|$ | 1.1 | 1.1 | 1.2 | 1.7 |
| $\frac{1}{n}\sum\|\sigma\|$ | 1.9 | 1.8 | 1.8 | 1.7 |

Pooled values exclude PF-1. $Md$=median difference from an individual site to the overall median, $\sigma$=standard deviation of measurements at a particular site, AM=air-mass-adjustment, T=temperature error adjustment, AK=averaging kernel adjustment, Trunc=125HR interferograms were truncated to yield reduced resolution spectra before comparison.

[Figure]

Figure 9. Medians and standard deviations of the TCCON data compared to the mFTS product after various adjustments. Line style represents the significance of the difference of the group median from the median of all data by the Kruskal-Wallis test. ($p < 0.05$ –, $p < 0.2$ --, otherwise ⋯). Legend entries indicate what adjustments were applied to the data to make measurements from the different instrument types more comparable. Open symbols did not have a scaling factor applied to center about zero. AM = adjustment with air-mass, T = temperature error adjustment, AK = averaging kernel adjustment.

[Figure]

Figure 10. Pairwise 95% CI of differences between sites. Differences for data within ±2 hours local noon. Comparisons are ranked in order of decreasing mean difference. For each species, plots are shown for 1) corrections with air mass, differences in temperature sensitivity, and a reduction of the smoothing error from different averaging kernels 2) differences by comparing results from 125HR spectra with lowered resolutions. At the bottom are the site orderings. Black lines between indicate when the pairwise difference is first more than 0.

**1.2**

**If the biases found between TCCON sites differ, we would assume that this difference is mainly due to ILS differences between sites. TCCON has implemented an instrumental line shape (ILS) monitoring based on calibrated gas cells: the authors could therefore check whether the empirical statistical findings are in agreement with the expected error propagation of an imperfect ILS - thereby closing the circle. Note that the foreseen capability of handling an imperfect ILS in the analysis is not required for this investigation - estimates of the sensitivity of TCCON data products with respect to modulation efficiency imperfections have been provided by e.g. D. Griffith and other investigators in the past. However, the exercise will provide a preview of the level of intercomparability which can be expected for TCCON when the ILS biases will be incorporated in the standard TCCON workflow (as announced by the authors).**

We have changed Sect. 4.4 from "Other Considerations" to "Effects of a non-ideal Instrument Line Shape." In this section we include results from ILS monitoring of the calibrated HCl gas cells (Fig. 7), and discuss sensitivity estimates from past studies for modulation efficiencies that are not unity. We also include discussion of using the $X_{air}$ parameter as a diagnostic for large misalignments, and include a comparison of $X_{air}$ in Fig. 8. The text of Sect. 4.4 is copied below.

**4.4 Effects of a non-ideal Instrument Line Shape**

Imperfections in the instrument line shape (ILS) due to misalignment of the FTSs can also cause site biases. At the sites described in this study, weekly internal lamp measurements of the internal, calibrated HCl cells (Hase et al., 2013) are collected from the 125HR instruments. We use LINEFIT 14.5 (Hase et al., 1999) software on HCl lines from monthly-averaged spectra to characterize the ILS. For Park Falls spectra were averaged before and after realignment. In Fig. 7 are the modulation efficiency (ME) and phase error (PE) with OPD. An ME not equal to one can indicate instrument misalignment, which may be from shear, angular, or defocus misalignment.

Effects of different types of misalignment on ME are not independent (Toon et al., 2016). However, parameterizing changes in ME with OPD can be used to assess effects on $X_{gas}$ retrievals (Griffith et al., 2010; Velazco et al., 2016; Wunch et al., 2011, 2015). These previous studies have found that each 1% increase in ME at MOPD leads to a decrease on order of 0.04% in $X_{CO2}$, though the change does vary with SZA. For $X_{CH4}$, there is a decrease on order of 0.03–0.05% for a 1% increase in ME at MOPD. The cause of the change in ME with OPD can, however, also significantly influence results. For example, Wunch et al., (2015) noted significantly different results for the same change in ME when the cause is shear versus angular misalignment.

We estimate biases based on ME at MOPD values alone, compared with AFRC. Based on the LINEFIT analysis of the lamp spectra, we would expect a low $X_{CO2}$ bias of 0.02% for Caltech, a high bias of 0.05% for Lamont, and a high bias of 0.24% for Park Falls (prior to realignment). The results of our study are not consistent with this expectation. Only Park Falls is consistently in the right direction with a bias of ~0.18% before realignment. After realignment Park Falls $X_{CO2}$ was more in line with the other spectrometers, although based on the ME at MOPD results alone there should have been little change. The Park Falls ILS was much more symmetrical after re-alignment, as seen by the PE curve in the lower panel of Fig. 7 being much closer to zero. For $X_{CH4}$, both Park Falls and Lamont are biased in the

expected direction from Armstrong, and the Park Falls-1 bias is ~0.17%. However, the Lamont bias is greater than expected from the single value parameterization. A more complex parameterization of the ILS effect on $X_{gas}$ (e.g. using the full function of ME with OPD, accounting for SZA dependence) might reduce the expected versus observed mismatch.

The $X_{air}$ parameter from GGG can be used as a diagnostic for large misalignments, timing errors, and surface pressure errors. $X_{air}$ is calculated by dividing the sum of all non-water molecules based on the surface pressure by the retrieved column of dry air based on column $O_2$. $X_{air}$ should be close to 1.0 and not vary, though empirically it is approximately 2% lower due to spectroscopic errors for oxygen (Washenfelder et al., 2006). Wunch et al. (2015) showed an increase of about 0.3% in $X_{air}$ for a 1% increase in ME at MOPD due to shear misalignment, and the change due to angular misalignment was <0.03%. In Fig. 8 $X_{air}$ is shown for all the sites. At Park Falls $X_{air}$ was approximately 0.979 before and 0.983 after alignment, which could correspond to an ME increase of about 0.013 at MOPD from shear realignment. Though LINEFIT results do not show a significant increase in ME at MOPD after 11 September 2015, both $X_{CO2}$ and $X_{CH4}$ changed in the expected direction (decreased). Based on $X_{air}$, $X_{CO2}$ was expected to change by ~0.2 ppm (compared with ~0.08 ppm) and $X_{CH4}$ was expected to change by 0.7–1.2 ppb (compared with ~1.5 ppb). Residual differences may indicate measurement uncertainties.

[Figure]

Figure 7. Modulation efficiency and phase error for each of the 125HR instruments describe the ILS. Results are calculated from HCl lines using LINEFIT 14.5 on monthly averages of internal lamp spectra. For Caltech, 2 different months are shown and Park Falls is split before and after realignment.

[Figure]

Figure 8. TCCON $X_{air}$ compared with mFTS $X_{air}$ within ± 2 hours of local noon. The differences are scaled by 1.001 to be centered about zero. $X_{air}$ can be used as a diagnostic for misalignments, timing, or surface pressure errors.

**Minor comments**

**1.3**

**Page 12, line 1 ff: "Of these, only the last three can cause site-to-site bias". Is this true? If e.g. two TCCON sites are operated in different latitudes, then a comparison between datasets of actual values and TCCON data would reveal a systematic bias between the sites due to spurious air-mass dependence from spectroscopic issues (if we assume that the two sites systematically cover different ranges of solar elevation angles).**

We assumed solar elevation angle (SEA) error was minimized here due to the empirical SEA correction to reduce the diurnal variability (Wunch et al., 2011). However, as pointed out these corrections are not perfect and there is residual systematic error (especially due to latitude dependence of SEAs).

Five reasons $X_{gas}$ could differ among instruments were considered: 1) differences in averaging kernels, 2) differences in spurious air-mass-dependence from spectroscopy errors, 3) the a priori profile (e.g. temperature profile), 4) error in the measured surface pressure, and 5) instrument misalignments. Of these, the last four can cause site-to-site biases in the TCCON, and empirical adjustments to make the mFTS and TCCON datasets more comparable were made to the first three.

Recent work by Matthäus Kiel has worked on quantifying this residual error. The following has been added to Sect. 4.1.

Further, for estimates of bias we only use data within ±2 hours of local noon so that comparisons are over similar SZAs at all sites. This constrains comparison data to have an AM between 1.05 and 1.85 (site means between 1.10 and 1.46). Recent work has shown residual dependencies on AM that could cause a high bias of ~1 ppb $X_{CH4}$ between AM 1.10 and 1.46 (Matthaeus Kiel, personal communications).

**1.4**

**Page 12, line 26: The suggestion of using several mobile spectrometers for the intercomparison seems to imply a substantial effort (if the spectrometers are not collocated at the site for performing differential measurements anyway). I would envisage a different recommended standard procedure, especially for remote TCCON sites: the use of a single spectrometer, including a careful demonstration that no instrumental drift occurred (perform an intercomparison with respect to a reference - ideally, a TCCON spectrometer and one or several mobile spectrometers remaining there - before and after the campaign).**

There are other advantages to using multiple spectrometers including less data loss if one instrument unexpectedly stops working in the field, and a higher SNR. The procedure recommended here works if there is no change between mobile spectrometers before and after the field comparison, or if it is known when changes occurred and how they affect retrievals. Ideally, as was the case here, there will be no change. But if there is only one spectrometer and it is damaged during shipment, it could setback the campaign several months. We include the suggested procedure, but as a backup.

Ideally repeat campaigns will include multiple traveling mFTS instruments. Others may even consider taking three mFTS instruments so if there is a change from one, it would be noticeable by comparing with the other two. When collocated, 3+ EM27/SUN instruments can easily be operated by just one or two people. Multiple instruments also provide backup in case problems arise with one, and can increase the signal to noise ratio. As a backup strategy, one travelling mFTS can be taken in the field and compared with an mFTS left in a fixed location before and after the campaign. This second strategy is acceptable when there are no instrumental issues, or if it is known when and how issues affect $X_{gas}$ measurements.

**1.5**
**In this context, I would find it useful to discuss in more detail the level of stability of the participating mobile spectrometers as reaction to transport events.**

We mention standard deviations of the measurements between the two mFTS instruments.

These were determined by comparing TCCON data with simultaneously collected data from co-located portable spectrometers, which we have assumed to be internally precise over the duration of the campaign. This assumption is supported by standard deviations of only 0.15 ppm for $X_{CO2}$ and 1 ppb for $X_{CH4}$ for the 10-minute averaged differences between the two mFTS instruments over the campaign.

**1.6**
**(The LANL spectrometer has been operated site-by-site to the TCCON spectrometer located in Karlsruhe before overseas shipment - it would be interesting to include these observations as well.)**

While we agree that including more comparisons would be interesting, we leave it to a future studies. Based on personal communications with our colleagues (Matthias Frey, and Mahesh Sha) there may be a sufficient number of observations from sites in and around Europe to perform a similar analysis. There may also be sufficient observations to do the same for the Western Pacific region (e.g. Japan, Australia, and New Zealand). We leave such analyses to those most familiar with the data.

**Responses to referee #2**

**2.1**

**However, the manuscript mentions only the possible biases due to the non-ideal instrumental line shape (ILS) without going into further details. In my opinion, this is an important piece of information which would help in understanding the cause of the residual biases between the TCCON sites.**
**I suggest to include a plot (or to give values) showing the ILS parameters of the 4 TCCON sites. The sensitivity of the TCCON retrieval results with respect to the ILS parameter has been studied in the past by several investigators. Therefore, the authors could use this information and check if their findings are in agreement with the expected bias due to an imperfect ILS of the TCCON spectrometer.**

We agree that a more detailed discussion of effects of non-ideal ILS as well as plots of ILS parameters from the 4 TCCON instruments is warranted. This is now included as Sect. 4.4. See also response **1.2** above.

**2.2**

**As a result, I suggest performing an additional step of truncating the TCCON measurements to the resolution of the mFTSs and making the intercomparison study. This would create identical measurements performed by the two different types of spectrometers at the same spectral resolution. The inter-comparison of the Xgas retrieved from the two datasets would eliminate several differences which are present when comparing the mFTS retrieval results with the high resolution TCCON retrieval results, but will preserve other errors which are instrument and detector related. This study would therefore give a more exhaustive investigation of the residual bias and their possible causes.**

See response **1.1** above.

**2.3**

**Page 7 line 4-5: What is the motivation of using a different scaling factor than that of the TCCON Xgas calculation scaling factor?**

We have added an explanation to the text of why we include an additional scaling factor.

The mFTS data were scaled to match the TCCON product and center the difference about zero, by dividing by scaling factors of 0.9987 for $X_{CO2}$ and 1.0073 for $X_{CH4}$. These factors were based on the TCCON and mFTS data at all sites and were used in combination with the TCCON to aircraft bias correction (Wunch et al., 2015). An additional scaling factor is used because retrievals from lower-resolution spectra are biased compared to higher resolution spectra due to errors in a priori profiles, and spectroscopy errors (Gisi et al., 2012; Hedelius et al., 2016; Petri et al., 2012).

**2.4**

**Page 6 line 10: Reference missing, e.g. Petri et al. 2012, Gisi et al. 2012**

Added.

Because of different spectral resolutions between the TCCON instruments (0.02 cm$^{-1}$) and the travelling spectrometers (0.5 cm$^{-1}$), we anticipate that there may be systematic differences in their $X_{gas}$ retrievals (Gisi et al., 2012; Petri et al., 2012).

References:

Gisi, M., Hase, F., Dohe, S., Blumenstock, T., Simon, A. and Keens, A.: $X_{CO2}$-measurements with a tabletop FTS using solar absorption spectroscopy, Atmos. Meas. Tech., 5, 2969–2980, doi:10.5194/amt-5-2969-2012, 2012.

Hase, F., Blumenstock, T. and Paton-Walsh, C.: Analysis of the instrumental line shape of high-resolution fourier transform IR spectrometers with gas cell measurements and new retrieval software., Appl. Opt., 38, 3417–3422, doi:10.1364/AO.38.003417, 1999.

Hase, F., Drouin, B. J., Roehl, C. M., Toon, G. C., Wennberg, P. O., Wunch, D., Blumenstock, T., Desmet, F., Feist, D. G., Heikkinen, P., De Mazière, M., Rettinger, M., Robinson, J., Schneider, M., Sherlock, V., Sussmann, R., Té, Y., Warneke, T. and Weinzierl, C.: Calibration of sealed HCl cells used for TCCON instrumental line shape monitoring, Atmos. Meas. Tech., 6(12), 3527–3537, doi:10.5194/amt-6-3527-2013, 2013.

Hedelius, J. K., Viatte, C., Wunch, D., Roehl, C., Toon, G. C., Chen, J., Jones, T., Wofsy, S. C., Franklin, J. E., Parker, H., Dubey, M. K. and Wennberg, P. O.: Assessment of errors and biases in retrievals of $X_{CO2}$, $X_{CH4}$, $X_{CO}$, and $X_{N2O}$ from a 0.5 $cm^{-1}$ resolution solar viewing spectrometer, Atmos. Meas. Tech., 9, 3527–3546, doi:10.5194/amt-9-3527-2016, 2016.

Petri, C., Warneke, T., Jones, N., Ridder, T., Messerschmidt, J., Weinzierl, T., Geibel, M. and Notholt, J.: Remote sensing of $CO_2$ and $CH_4$ using solar absorption spectrometry with a low resolution spectrometer, Atmos. Meas. Tech., 5, 1627–1635, doi:10.5194/amt-5-1627-2012, 2012.

Washenfelder, R. A., Toon, G. C., Blavier, J. F., Yang, Z., Allen, N. T., Wennberg, P. O., Vay, S. A., Matross, D. M. and Daube, B. C.: Carbon dioxide column abundances at the Wisconsin Tall Tower site, J. Geophys. Res. Atmos., 111, D22305, doi:10.1029/2006JD007154, 2006.

Wunch, D., Toon, G. C., Blavier, J.-F. L., Washenfelder, R. A., Notholt, J., Connor, B. J., Griffith, D. W. T., Sherlock, V. and Wennberg, P. O.: The Total Carbon Column Observing Network, Philos. Trans. R. Soc. A, 369, 2087–2112, doi:10.1098/rsta.2010.0240, 2011.

Wunch, D., Toon, G. C., Sherlock, V., Deutscher, N. M., Liu, C., Feist, D. G. and Wennberg, P. O.: The Total Carbon Column Observing Network's GGG2014 Data Version, 43, doi:10.14291/tccon.ggg2014.documentation.R0/1221662, 2015.